# Language-Specific Latent Process Hinders Cross-Lingual Performance

## Abstract

Large language models (LLMs) are demonstrably capable of cross-lingual transfer, but can produce inconsistent output when prompted with the same queries written in different languages. To understand how language models are able to generalize knowledge from one language to the others, we measure representation similarity between languages by centered kernel alignment (CKA) and cosine similarity. We also apply the *logit lens* to interpret the implicit steps taken by LLMs to solve multilingual multi-choice reasoning questions. Our analyses reveal LLMs predict inconsistently and are less accurate because they rely on representations that are dissimilar across languages, rather than working in a shared semantic space. While larger models are more multilingual, we show their hidden states are more likely to dissociate from the shared representation compared to smaller models, but are nevertheless more capable of retrieving knowledge embedded across different languages. Finally, we demonstrate that knowledge sharing in small models can be facilitated by steering their latent processing towards the shared semantic space. This improves the models' multilingual reasoning performance, as a result of more knowledge transfer from, and better output consistency with English.

## 1 Introduction

Humans have an innate ability to apply common knowledge and perform reasoning skills consistently across different languages, as the cognitive functions involved are largely independent of the languages they are learned in (Fodor, 1983; Dehaene, 2011). Modern large language models (LLMs) are trained predominantly in English (Rivière et al., 2024; Grattafiori et al., 2024) and Chinese (Yang et al., 2024a). To some degree, LLMs demonstrate cross-lingual capabilities (Shi et al., 2023), but the performance gap between languages remains (Kassner et al., 2021; Jiang et al., 2020), and the models struggle to maintain consistent responses when they are probed by the same queries in different languages (Qi et al., 2023; Ifergan et al., 2024; Goldman et al., 2025).

One possible explanation is that language models tend to develop language-specific neurons, rather than sharing them among multiple languages (Tang et al., 2024). Nevertheless, language-specific neurons have never been shown to cause inconsistency in LLMs directly. Other work suggests that LLMs operate in a shared subspace closest to the English tokens, before projecting back to other languages in higher layers, for predictions in multilingual and multi-modal tasks (Wendler et al., 2024; Wu et al., 2024; Zhao et al., 2024b).[1] These findings, however, fail to explain the lack of consistency across languages for the largest models (Goldman et al., 2025) — otherwise, model outputs would converge toward English-prompted responses as multilingual performance improves.

This work investigates how language models generalize knowledge to multiple languages. We experiment on Gemma 2 and Qwen 2.5 models and show that:

**Utilization of shared semantic space in multilingual processing facilitates knowledge transfer.** We first evaluate LLMs' cross-lingual consistency and transfer on multi-choice queries. We find larger LLMs are indeed more accurate, but present large performance gaps, and remain inconsistent with subpar cross-lingual transfer. Through measuring cross-lingual alignment in representations, we

---

[1] Wang et al. (2025a), for example, attribute language inconsistency to the models' failure to transition from English to the target language at the final layers.

discover that models produce more divergent responses when their internal representations are more dissimilar across languages. We present evidence demonstrating a strong link between utilization of a shared semantic space and the transfer performance of LLMs.

**Nonetheless, larger models tend to reason in the native language of the prompt.** Despite their superior multilingual performance, we find representations in larger LLMs to be more structurally dissimilar and farther apart in the semantic space across languages. Further analyses with logit lens reveal that the representations move towards individual subspaces of the native languages. In contrast, smaller models tend to work predominantly in the English shared space. We attribute this to larger models' capability of encoding and retrieving information in different languages independently with significantly more parameters.

**Inducing processing in the shared semantic space improves cross-lingual transfer and performance in smaller models.** To provide a causal evidence for our claims, we apply cross-lingual steering in the final part of this work, inspired by Turner et al. (2023); Rimsky et al. (2024). These linear steering vectors are constructed to reinforce latent processing in the shared space, leading to more English-like predictions and performance gains. We find adding a linear steering vector towards English is effective for small models, but not for larger ones, likely due to their increased non-linearity and structurally distinct language subspaces.

In summary, our work provides an account of the internal multilingual processing of LLMs and explains the limitations of current models in knowledge transfer. To further validate our hypotheses we explore cross-lingual activation steering. We recommend future work to develop more comprehensive methods that encode knowledge of different languages in the shared semantic space to maximize transfer between languages.

## 2 METHODOLOGY

### 2.1 CROSS-LINGUAL CONSISTENCY AND KNOWLEDGE TRANSFER

We define an LLM to be cross-lingually consistent when it produces the same output to the same queries written in different languages. Unlike Ifergan et al. (2024), our measure of consistency is based upon multiple-choice question-answering (MCQ) tasks. To reduce language variability in the output space, we formulate the consistency between a language pair by their similarity in answer ranks, given by an LLM when prompted with the same questions and answer choices in their respective languages. MCQ ranks are informative, as the distractors are typically selected to align to feasible mistakes, which we expect to transfer across languages.

Formally, for each of $i \in I$ questions in language $l$, $q_{i,l} \in Q_l$, paired with $j \in \{1..J\}$ answer choices $[a_{i,l}^1, ..., a_{i,l}^J]$, we rank the answer choices by their output probability $p(a_{i,l}^j | q_{i,l})$ given the question. We then combine the ranks $k(a_{i,l}^j)$ of each answer within the set of choices:

$$\mathbf{k}_l = \langle k(a_{1,l}^1), ..., k(a_{1,l}^J), k(a_{2,l}^1), ..., k(a_{I,l}^J) \rangle.$$

Thus, the pairwise consistency between languages $l_1$ and $l_2$ is defined as the Spearman's rank correlation coefficient between the rank vectors:[2]

$$\text{consistency}(l_1, l_2) = \rho(\mathbf{k}_{l_1}, \mathbf{k}_{l_2}). \tag{1}$$

Not all transfer between languages is beneficial—a model can be cross-lingually consistent but make many incorrect predictions. To approximate the rate of correct answers in a language propagates to another, we adapt our approach from Ifergan et al. (2024) and estimate the positive transfer from $l_1$ to $l_2$ by the ratio of shared accurate responses to all accurate responses in $l_1$:

$$\text{tr}^+(l_1, l_2) = |F_{l_1} \cap F_{l_2}| / |F_{l_1}|. \tag{2}$$

---

[2]We choose Spearman's correlation based on ranks over Pearson's correlation based on answer distributions because the former is less sensitive to variance introduced by multilingual textual outputs. We decide against KL divergence for similar reason, and because the concept of consistency is more intuitive than inconsistency.

$F_l$ denotes the set of questions in $l$ that the model answers correctly. Conversely, $\neg F_l$ refers to the set of questions in $l$ where the model responses incorrectly, and $\neg F_{l \cap m}$ the questions in $l$ and $m$ where the model return the *same* incorrect responses. We compute the negative transfer from $l_1$ as:

$$\mathrm{tr}^-(l_1, l_2) = |\neg F_{l_1 \cap l_2}| / |\neg F_{l_1}|. \tag{3}$$

We note that transfer rates defined in Equations 2 and 3 are directed and non-commutative, unlike consistency in Equation 1. Further, it is not possible to satisfy all the listed requirements without perfect accuracy—full consistency is not attainable without both maximum positive and negative transfer rates. Overall, the consistency, positive and negative transfer rates of a model can be computed by the expected values for all pairwise combinations of languages, $N = \{l_1, l_2 \in L \mid l_1 \neq l_2\}$:

$$\mathbb{E}[\text{consistency}] = \frac{1}{|N|} \sum_{l_1, l_2 \in N} \text{consistency}(l_1, l_2), \tag{4}$$

$$\mathbb{E}[\text{positive transfer}] = \frac{1}{|N|} \sum_{l_1, l_2 \in N} \mathrm{tr}^+(l_1, l_2), \tag{5}$$

$$\mathbb{E}[\text{negative transfer}] = \frac{1}{|N|} \sum_{l_1, l_2 \in N} \mathrm{tr}^-(l_1, l_2). \tag{6}$$

## 2.2 Cross-lingual alignment and multilingual processing in the latent space

Like Hämmerl et al. (2024), we hold the view that cross-lingual alignment is a property where models represent queries in different languages with equivalent semantics more similarly than those with dissimilar semantics. We choose two metrics to quantify cross-lingual similarity in representations: linear CKA and cosine similarity, which we justify below. To further provide insights on the ways in which representations are dissimilar, we follow prior work and use logit lens, which estimate the similarity between the hidden representation and the language-specific embedding space (Wendler et al., 2024; Alabi et al., 2024).

**Linear CKA.** CKA is proposed as a similarity measure between representations and is invariant to any orthogonal transformations and isotropic scaling (Kornblith et al., 2019). These properties make them ideal for determining the structural similarity between neural networks that have different random initializations and widths. We argue that CKA could also be useful for detecting any meaningful similarity between parallel representations of different languages and for comparison across models at different scales. Following Kornblith et al. (2019), we apply a linear kernel for all our CKA measures, unless otherwise specified.

Formally, let $X = [h_{1,\ell}^{l_1}, ..., h_{n,\ell}^{l_1}]$, $Y = [h_{1,\ell}^{l_2}, ..., h_{n,\ell}^{l_2}]$ be the representations of $n$ parallel queries in languages $l_1$ and $l_2$ for layer $\ell$. The linear CKA between languages is defined as

$$\mathrm{CKA}_{\mathrm{linear}}(l_1, l_2, \ell) = \frac{\mathrm{tr}(X^T Y \cdot Y^T X)}{\|XX^T\|_{\mathrm{F}} \|YY^T\|_{\mathrm{F}}}. \tag{7}$$

**Cosine similarity.** Cosine similarity is a widely used metric for cross-lingual alignment (Hämmerl et al., 2024; Chua et al., 2024; Wang et al., 2025a). We compute the cosine similarity between a language pair by the mean similarity over a set of parallel queries:

$$\cos(l_1, l_2, \ell) = \frac{1}{n} \sum_{i=1}^{n} \frac{h_{i,\ell}^{l_1} \cdot h_{i,\ell}^{l_2}}{\|h_{i,\ell}^{l_1}\| \|h_{i,\ell}^{l_2}\|}. \tag{8}$$

The function captures the degree to which two representations belong in the same neighborhood, but is insensitive to the structural relations between examples (Linzen, 2016; Zhou et al., 2022), unlike CKA. Further, cosine similarity inherently favors high-dimensional space, which is undesirable for our purpose of comparing between smaller and larger models. To mitigate this effect, we take inspiration from Hämmerl et al. (2024) and compute language-pair similarity as its ratio to the similarity between representations of different semantics in the monolingual space of either language.

This can be obtained by repeated sampling from the same query set, provided that the paired samples are different to each other:

$$\cos(l_1, \ell) = \frac{1}{n} \sum_{i=1, j \neq i}^{n} \frac{h_{i,\ell}^{l_1} \cdot h_{j,\ell}^{l_1}}{\|h_{i,\ell}^{l_1}\| \|h_{j,\ell}^{l_1}\|}. \tag{9}$$

Finally, we define the normalized cosine similarity between languages as the harmonic mean of the normalizations by $\cos(l_1, \ell)$ and $\cos(l_2, \ell)$:

$$\cos_{\text{norm}}(l_1, l_2, \ell) = \frac{2\cos_{l_1}\cos_{l_2}}{\cos_{l_1} + \cos_{l_2}}, \tag{10}$$

$$\text{where} \quad \cos_{l_1} = \cos(l_1, l_2, \ell)/\cos(l_1, \ell) \quad \text{and} \quad \cos_{l_2} = \cos(l_1, l_2, \ell)/\cos(l_2, \ell). \tag{11}$$

**Logit lens.** To characterize the cross-lingual differences in representation, we rely on the *logit lens* to interpret the hidden states by applying the output *softmax* layer to the vectors (Nostalgebraist, 2020). These intermediate distributions have been found to provide insights and largely explain the final predictions of a model (Wendler et al., 2024; Alabi et al., 2024; Wu et al., 2024).

Logit lens is primarily designed to probe the latent distribution of the next token. To estimate the probability of a sequence in the latent space, we follow Zhong et al. (2024) to allow past tokens to be included in the input, and decode from the hidden states to compute the final probabilities at each layer. Let $[x_t, ..., x_T]$ be the token indices of a phrase $\mathbf{w}$ following a query $\mathbf{q}$ with token indices $[x_0, ..., x_{t-1}]$, $h_\ell^t$ be the hidden state of layer $\ell$ at timestep $t$, and $U$ denote the unembedding matrix. The latent probability of $\mathbf{w}$ at layer $\ell$, $p_\ell(\mathbf{w}|\mathbf{q})$, is equivalent to

$$\prod_{\tau \in [t..T]} \text{softmax}(U h_\ell^t)_{x_\tau} \tag{12}$$

which we normalize by the length of $\mathbf{w}$: $\sqrt[T-t+1]{p_\ell(\mathbf{w}|\mathbf{q})}$.[3]

Based on these equations, we operationalize the relative similarity between the hidden states and embedding space in terms of the normalized probabilities of the answer choices in full textual forms. For all queries, we compute the probabilities of the answer choices in the native language, as well as that of English parallel choices. Using the latent probabilities, we test if language models could reason in languages beyond English and extract information stored in different languages (Wendler et al., 2024; Alabi et al., 2024; Zhao et al., 2024b; Wu et al., 2024; Ifergan et al., 2024).

Cross-lingual alignment is assumed to underlie zero-shot cross-lingual transfer for many applications (Hämmerl et al., 2024). Therefore, our first hypothesis is that languages with more similar representations to those of other languages will also be more consistent. Based on Tang et al. (2024) and Goldman et al. (2025), we also hypothesize that large-scale LLMs, with a greater number of language-specific neurons in the hidden layers, could induce more diverging representations across languages. Overall, we expect models that are more cross-lingually aligned to operate in the shared semantic space, which is closest to English, rather than individual language subspace, facilitating cross-lingual knowledge transfer and consistency.

## 2.3 CROSS-LINGUAL ACTIVATION STEERING

To further test these hypotheses, we derive from model behavior steering literature (Turner et al., 2023; Rimsky et al., 2024) and propose cross-lingual activation steering. Because the shared latent space is most similar to English (Wendler et al., 2024), we expect pushing the latent processes towards English to induce processing in the shared space, allowing for knowledge sharing, whereas reversing the process will cause more divergent predictions.

Following Rimsky et al. (2024), we obtain the contrastive addition vector of multiple-choice questions for each languages, by computing the mean difference in activation as the model processes parallel

---

[3]The approach is analogous to prior work that analyzes the latent representation of LLMs, where traces of reasoning (Yang et al., 2024b) and abstract concepts (Jin et al., 2025) are found, providing insights into model predictions. The proposed method is also plausible because it is directly related to how an autoregressive model predicts the next token, where a textual answer can be sampled from the token distribution.

Table 1: Mean accuracy (↑), consistency (↑), positive (↑) and negative (↓) transfer. Larger models are more accurate, but with larger performance gap and limited transfer across languages.

| | BELEBELE | | | | mMMLU | | | | mHellaSwag | | | |
|---|---|---|---|---|---|---|---|---|---|---|---|---|
| | acc. | cons. | tr$^+$ | tr$^-$ | acc. | cons. | tr$^+$ | tr$^-$ | acc. | cons. | tr$^+$ | tr$^-$ |
| Gemma 2 2B | 47.1±13.8 | .381 | .623 | .372 | 42.5±5.9 | .540 | .671 | .509 | 44.8±8.3 | .461 | .656 | .470 |
| Gemma 2 9B | 72.4±18.4 | .427 | .779 | .273 | 59.0±8.2 | .561 | .771 | .478 | 66.2±11.9 | .558 | .780 | .418 |
| Gemma 2 27B | 75.1±18.2 | .548 | .803 | .289 | 62.6±8.3 | .639 | .810 | .521 | 69.9±12.3 | .631 | .807 | .429 |
| Qwen 2.5 3B | 51.7±17.9 | .276 | .599 | .278 | 47.8±9.1 | .423 | .662 | .440 | 52.9±14.1 | .418 | .663 | .385 |
| Qwen 2.5 7B | 57.6±20.3 | .383 | .650 | .278 | 54.9±10.6 | .541 | .724 | .442 | 60.4±15.0 | .474 | .718 | .381 |
| Qwen 2.5 32B | 70.3±20.9 | .485 | .757 | .277 | 66.4±11.8 | .567 | .798 | .443 | 70.0±14.2 | .555 | .796 | .396 |

queries in English and each of the languages, for each data set. The vector is extracted from the last token of the prompts, as the models prepare to generate an answer in the next token. Specifically, for a set of parallel queries in English and language $m$, $Q_{en,m}$, the steering vector at layer $\ell$ from $m$ to English is computed as:

$$v_{m \to en}^\ell = \frac{1}{|Q_{en,m}|} \sum_{q_{en}, q_m \in Q_{en,m}} h_\ell(q_{en}) - h_\ell(q_m), \tag{13}$$

where $h_\ell(...)$ refers to the hidden activation at layer $\ell$ of the query's last token. Like previous work (Rimsky et al., 2024), we use a scalar multiplier $\gamma$ to modulate the direction of change and the magnitude of the vectors. $\gamma > 0$ induces latent process similar to being prompted in English. At inference, the extracted addition vector is added to the hidden activation as $\gamma \cdot v_{m \to en}^\ell$ at the last token of the prompts.

## 3 EXPERIMENTAL SETUP

**Models.** We experiment on Gemma 2 (Rivière et al., 2024) and Qwen 2.5 (Yang et al., 2024a) family of language models, which have been trained and instruction-tuned primarily in English. We use 2B, 9B and 27B variants of Gemma models and 3B, 7B and 32B of Qwen models. Note that the Qwen models have been tuned on multilingual instructions and are claimed to support 29 languages (Qwen Team, 2024). For all models, we compute the output distribution of the hidden states at an interval of four layers, ending at the final layer (e.g., layer 2, 6, ..., 42; where 42 is the total number of layers in Gemma 2 9B).

**Data.** We examine the latent process of the models using zero-shot prompts with three parallel multilingual tasks: BELEBELE for machine reading comprehension in 122 languages, created by human translators (Bandarkar et al., 2023); mMMLU for world knowledge and problem solving questions that have been manually translated and are available in 15 languages (Hendrycks et al., 2020; OpenAI, 2024); and mHellaSwag for commonsense natural language inference in 10 languages, created with automatic translations (Lai et al., 2023; Zellers et al., 2019).

For multiple-choice questions, we associate each answer choice with an alphabetic characters (e.g., 'A'). The model output is constrained to these characters to ensure any observations made in the latent space are not due to variance in multilingual output.[4] We then evaluate the models based on the output distribution of the alphabets listed in the prompts. All prompt templates used are reported in Appendix A. We follow Bandarkar et al. (2023) to compute the accuracy of the models in these tasks.

## 4 RESULTS

We report the accuracy and cross-lingual transfer performance of all models based on the ranked alphabet choices. Table 1 shows the mean accuracy across languages and the consistency and transfer

---

[4]With our setup, it is possible for models to achieve perfect consistency with maximally aligned representations, and any variation we observe can solely be attributed to the differences in multilingual processing.

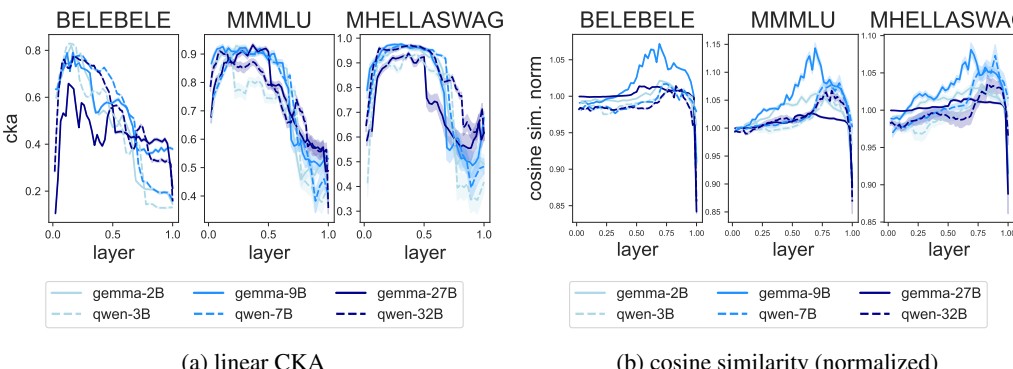

(a) linear CKA       (b) cosine similarity (normalized)

Figure 1: Mean similarity between representations of language pairs. Latent representations of Gemma 9B and Qwen 7B are the most aligned cross-lingually relative to their smaller and larger counterparts (lighter and darker blues respectively), especially towards the middle layers. Shaded areas indicate 95% confidence intervals across language pairs. Note that each model consists of different number of layers, which we normalize by the maximum layer along x-axes.

Table 2: Pearson's correlations between averaged cross-lingual similarity measures of languages and their output accuracy, consistency and positive transfer from other languages. * indicates $p < .05$, ** indicates $p < .01$ and ***, $p < .001$.

| | BELEBELE | | | mMMLU | | | mHellaSwag | | |
|---|---|---|---|---|---|---|---|---|---|
| | acc. | cons. | →tr$^+$ | acc. | cons. | →tr$^+$ | acc. | cons. | →tr$^+$ |
| *linear CKA* | | | | | | | | | |
| Gemma 2 2B | $0.76^{***}$ | $0.85^{***}$ | $0.81^{***}$ | $0.64^{**}$ | $0.95^{***}$ | $0.81^{***}$ | $0.72^{*}$ | $0.90^{***}$ | $0.88^{***}$ |
| Gemma 2 9B | $0.94^{***}$ | $0.95^{***}$ | $0.94^{***}$ | $0.94^{***}$ | $0.99^{***}$ | $0.97^{***}$ | $0.93^{***}$ | $0.99^{***}$ | $0.97^{***}$ |
| Gemma 2 27B | $0.97^{***}$ | $0.97^{***}$ | $0.97^{***}$ | $0.95^{***}$ | $0.98^{***}$ | $0.97^{***}$ | $0.90^{***}$ | $0.97^{***}$ | $0.95^{***}$ |
| Qwen 2.5 3B | $0.44^{***}$ | $0.56^{***}$ | $0.45^{***}$ | $0.79^{***}$ | $0.86^{***}$ | $0.83^{***}$ | $0.84^{**}$ | $0.95^{***}$ | $0.89^{***}$ |
| Qwen 2.5 7B | $0.61^{***}$ | $0.60^{***}$ | $0.63^{***}$ | $0.86^{***}$ | $0.89^{***}$ | $0.87^{***}$ | $0.92^{***}$ | $0.99^{***}$ | $0.96^{***}$ |
| Qwen 2.5 32B | $0.74^{***}$ | $0.75^{***}$ | $0.75^{***}$ | $0.68^{**}$ | $0.71^{**}$ | $0.71^{**}$ | $0.94^{***}$ | $0.96^{***}$ | $0.96^{***}$ |
| *cosine similarity (normalized)* | | | | | | | | | |
| Gemma 2 2B | $0.68^{***}$ | $0.82^{***}$ | $0.74^{***}$ | $0.76^{***}$ | $0.96^{***}$ | $0.90^{***}$ | $0.69^{*}$ | $0.88^{***}$ | $0.85^{**}$ |
| Gemma 2 9B | $0.93^{***}$ | $0.96^{***}$ | $0.94^{***}$ | $0.90^{***}$ | $0.97^{***}$ | $0.94^{***}$ | $0.91^{***}$ | $0.98^{***}$ | $0.96^{***}$ |
| Gemma 2 27B | $0.78^{***}$ | $0.81^{***}$ | $0.79^{***}$ | $0.87^{***}$ | $0.85^{***}$ | $0.87^{***}$ | $0.92^{***}$ | $0.97^{***}$ | $0.96^{***}$ |
| Qwen 2.5 3B | $0.33^{***}$ | $0.34^{***}$ | $0.34^{***}$ | $0.47$ | $0.66^{**}$ | $0.56^{*}$ | $0.51$ | $0.65^{*}$ | $0.57$ |
| Qwen 2.5 7B | $0.56^{***}$ | $0.65^{***}$ | $0.58^{***}$ | $0.84^{***}$ | $0.89^{***}$ | $0.86^{***}$ | $0.82^{**}$ | $0.93^{***}$ | $0.88^{***}$ |
| Qwen 2.5 32B | $0.33^{***}$ | $0.36^{***}$ | $0.34^{***}$ | $0.63^{*}$ | $0.62^{*}$ | $0.65^{**}$ | $0.73^{*}$ | $0.81^{**}$ | $0.78^{**}$ |

rates (Equations 4, 5 and 6) for individual language models and benchmarks. Despite being more accurate, larger models such as Gemma 2 27B and Qwen 2.5 32B have more varied performances. In terms of consistency and transfer across languages, larger models offer little gain compared to their scale and accuracy, especially relative to the smaller models. Like Ifergan et al. (2024), we also found the impact of language relations on cross-lingual transfer, which we discuss in Appendix C.

## 4.1 CROSS-LINGUALLY SIMILAR REPRESENTATIONS PREDICT KNOWLEDGE TRANSFER

We compute cross-lingual alignment by CKA and cosine similarity (Equations 7 and 10), based on 50 randomly sampled, parallel queries from each data set. Figure 1 shows the similarity values across layers for all models. CKA reaches the peak values in the lower layers then decreases. In contrast, cosine similarity in the early layers suggests little alignment, but continues to increase at higher layers (< 0.8). This indicates multilingual inputs are initially encoded in a structurally similar way, but are far apart in latent space. In deeper layers, however, the representations move towards the

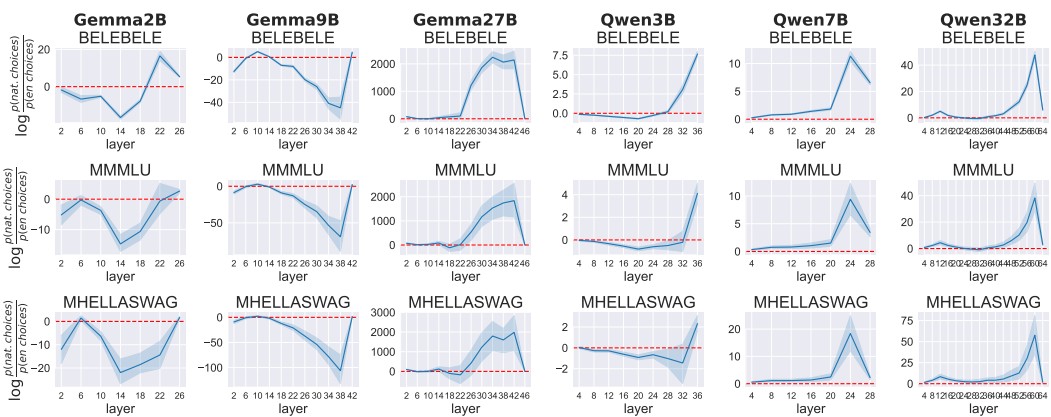

(a) Log ratio of latent probabilities of answer choices in the native language and English. Models assign more mass to the native language and operate in the language subspace when the log ratio exceeds zero.

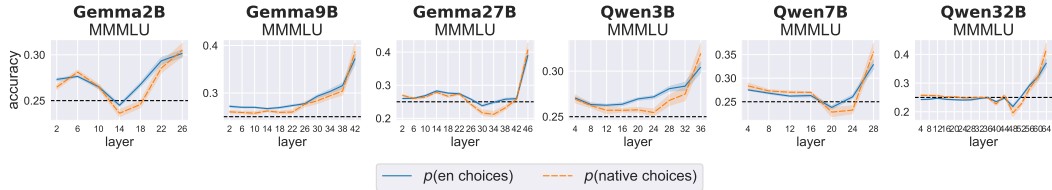

(b) Mean accuracy in the latent space across languages for mMMLU. Correct answers can be partially identified in the hidden states. Chance-level accuracy is marked with a dashed line. Full results with the other data sets are reported in Appendix D.

Figure 2: Mean log ratio of latent probabilities and latent accuracy across languages. Larger models tend to operate in a targeted subspace of individual languages, rather than the shared semantic space that is dominant in smaller models. English representation appears more stable and accurate. The shaded areas are 95% confidence intervals across languages.

same direction, but become less similar in terms of geometric structure. Nevertheless, we observe in both cases that middle-sized models (Gemma 9B and Qwen 7B) are more cross-lingually aligned than the smaller and larger models. This partially supports our hypothesis, and occurs in spite of the largest models (Gemma 27B and Qwen 32B) attaining the best performance in multilingual tasks. It is thus possible that the models could encode and retrieve more accurately knowledge learned in different languages independently with higher capacity. Representations then diverge closest to the output layer, as the models prepare to generate the next token in the native language.

Do models predict more accurately and consistently with shared representation? Table 2 reports the Pearson's correlations between the similarity measures (averaged across layers) extracted from 50 examples, and accuracy, consistency and positive transfer across languages of held-out examples. $\rightarrow tr^+$ refers to the mean positive transfer from all source languages (Equation 2). Overall, the results demonstrate strong predictive power of the similarity meausres on transfer performance, and that languages with more similar representations to the others are more likely to receive positive transfer, make more consistent predictions, and are therefore, more accurate.

## 4.2 MULTILINGUAL PROCESSING IN THE LATENT SPACE

Our next analyses explore multilingual processing with the normalized latent probabilities of answer choices in their native form (`nat.`) and their English equivalents (`en`); based on Equation 12.

Figure 2a shows the log ratios between $p(\text{nat. choices})$ and $p(\text{en choices})$, and reveals that larger models tend to assign more mass to answers in the native form in the intermediate layers (ratio above zero). At layer 22 and beyond, the probability of the native language exceed that of English in Gemma 27B. We see a similar trend in Qwen 7B and 32B with English at layers 4 and 40 respectively. This indicates that large-scale language models operate in individual space that is closer to the token

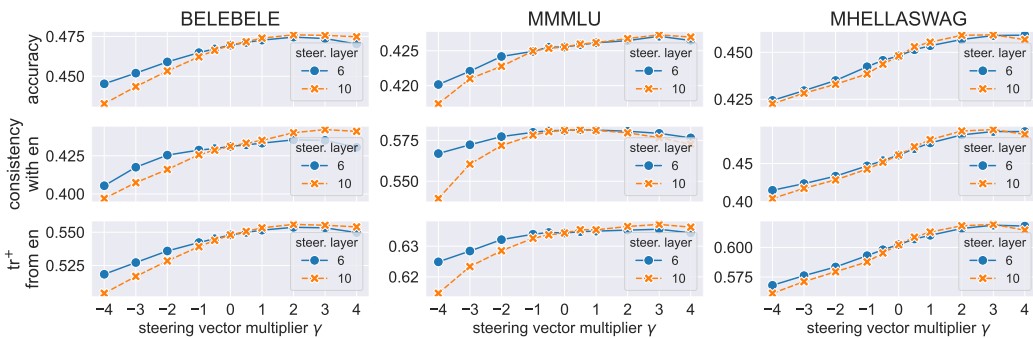

Figure 3: Steered Gemma 2B. Note that $\gamma > 0$ indicates an activation vector that pushes the model towards English processing in the latent space.

embeddings of the native language. The opposite behavior is observed for smaller models: Gemma 2B/9B and Qwen 3B. Early layers of the models are similar to the embeddings of the native languages. The hidden states then shift to working predominantly in the English in the middle layers, where the ratio is below zero, before moving closer to the native language space again at the highest layers. This is in line with previous findings (Tang et al., 2024; Zhao et al., 2024b) that most language-specific neurons are developed at the top and bottom layers of LLMs.

We further interpret the implicit steps taken across layers, by computing for all languages the accuracy of the native and English choices in the latent space. In Figure 2b (and Figure 5, Appendix D), the latent predictions are shown to be noisy, however, exhibit accuracy above chance level in higher layers across data sets. Nonetheless, English latent predictions appear more accurate and stable in most cases.

Together with the results in Section 4.1, our analyses imply that smaller models are limited to working within a shared representation, that is closest to English. Conversely, larger models transform the hidden states into subspace of individual languages, rather than the dominant representation. While it is possible for models to reason in a particular language, the process also hinders the models' ability to infer from the other languages, contributing to cross-lingual inconsistency.

## 4.3 Modulating cross-lingual consistency with activation steering

To establish a causal relation between English-aligned representation and transfer performance, we extract steering vectors from the 50 randomly sampled parallel questions from each data set, and evaluate the cross-lingual performance of the steered models on held-out data.[56]

Figure 3 reports a sweep of the steering vector multiplier, $\gamma$, on layers 6 and 10 of Gemma 2B. Positively steered models become increasingly accurate (peaking at $2 \leq \gamma \leq 3$), are more consistent with English and receive more positive transfer from English.[7]

We also observe that steering vectors at the middle layers are the most impactful, because the hidden representations are intended to be aligned at this stage (Figure 6, Appendix E), to allow for language-agnostic task solving (Zhao et al., 2024b).[8] In Appendix F.3, Figure 10 shows the accuracy changes (%) in steered models across languages. In general, low-resource (accuracy) languages written in Latin script benefit the most from the method in BELEBELE; whereas the improvement in mMMLU and mHellaSwag is less notable without steering vectors specialized to each topics. The method is also less effective on larger models. It is possible that the hidden representations become increasingly

---

[5]We reused the training set from Section 4.1.

[6]Prior to steering models, we perform PCA (Wold et al., 1987) to test if the representations are linearly separable, which predetermines the efficacy of the method. We document the results in Appendix E.

[7]We observe generally consistent variability across $\gamma$, in terms of standard deviations and confidence intervals over languages. This is omitted in the graph to avoid visual clutter. Similar plots for Qwen models are included in Appendix F.1.

[8]The performance of steered Gemma models across layers is reported in Figure 9 in Appendix F.2.

non-linear with scale, leading to more structurally different semantic spaces that are farther apart, as we have shown in Figure 1. Linear perturbations with steering vectors might therefore induce noise and break the geometric structure in the target space of larger models.

## 5 RELATED WORK

**Cross-lingual consistency and knowledge sharing.** Qi et al. (2023) define pairwise cross-lingual knowledge consistency as the equality in ranked outputs scaled by softmax, whereas Ifergan et al. (2024) and Goldman et al. (2025) measure the extent to which a language pair are accurate in a similar way with factual questions. Our use of Spearman's correlation in the evaluation of models' consistency allows a more direct approach, while also taking into account the relative order of ranks. Across languages, script similarity has been proposed to be a major factor to drive knowledge transfer (Qi et al., 2023; Ifergan et al., 2024; Goldman et al., 2025). Our work also supports previous findings that models are capable of processing and storing information with independent representation of different languages (Ifergan et al., 2024), and explains previous findings that transfer performance increases with model size only up to 14B for Qwen 2.5, indicating the ability of larger models to retrieve knowledge learned from independent languages (Goldman et al., 2025).

**Cross-lingual representation alignment.** A growing body of work has shown that multilingual LLMs use English as a pivot language in the hidden layers (Wendler et al., 2024), and share the subspace across languages and modalities (Wu et al., 2024). Alabi et al. (2024) find that the phenomenon also applies to models adapted to new languages, where finetuned adapters preserve most of the representation of the source language of the model, rather than working in a parallel isolated structure. Other works suggest the distinction between language-agnostic and language-specific neurons (Zhao et al., 2024b; Wang et al., 2024), motivating a series of methods that improve multilingual representation and performance of LLMs with targeted supervision (Zhao et al., 2024a; Huo et al., 2025; Wang et al., 2025b). The current study extends this understanding of multilingual models, and provides evidence for larger LLM's capability to encode meaningful representation in languages beyond English.

**English-centric multilingual reasoning.** Prior work has shown that explicit alignment with English reasoning improves multilingual performance. Existing approaches include English translations of multilingual queries (Shi et al., 2023; Zhu et al., 2024; Ko et al., 2025), using English reasoning traces (Shi et al., 2023; Qin et al., 2023; Yong et al., 2025; Ko et al., 2025), post-hoc translation from English answers (Ko et al., 2025), and trained alignment with English representation (Yoon et al., 2024; Huang et al., 2024; She et al., 2024). Our exploration on cross-lingual activation steering adds to this line of work, and offers a straightforward method to tap into English reasoning capability without training or additional inference costs.

## 6 DISCUSSION

This work investigates the challenges of cross-lingual transfer and consistency in LLMs. We examine the implicit processing of multilingual queries and show languages with diverging representations from the shared semantic space exhibit less consistency and knowledge transfer. With more capacity, larger models show a higher utilization of independent subspaces of individual languages, whereas smaller models are constrained to work predominantly in the shared space. Our work also demonstrates that reinforcing processing in the shared space via activation engineering improves multilingual reasoning performance and knowledge transfer in smaller models.

Despite the contributions of this study, we address several limitations. While there is a noticeable gain in performance with steered models, we only apply steering to establish causal evidence for our hypothesis. The method is not meant to be a practical alignment method, as it is only tested on multi-choice questions, and requires extraction of specific activation vectors for individual languages, which may not generalize to other use cases. We also caution against the use of English-centric methods beyond reasoning tasks, which might reinforce cultural and linguistic biases in LLMs (Singh et al., 2024; Rystrøm et al., 2025; Yong et al., 2025). It is however possible for future approaches to learn more stable internal representations when processing multilingual queries, which allow for complete retrieval of knowledge learned in different languages.

ETHICS STATEMENT

We use steering for the specific purpose of testing the causal relationship proposed by our hypotheses. A significant limitation of this method is its potential to reinforce existing bias towards English in LLMs, which may lead to unfair or harmful outcomes for non-English speakers. We thus recommend future research to address this limitation and to contribute to more equitable multilingual models.

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

# A   PROMPT TEMPLATE

Table 3: Prompt templates used for BELEBELE, mMMMLU and mHellaSwag.

---

**BELEBELE**

*Given the following question and answer choices, output the letter corresponding to the correct answer.*
`<passage>`
*Question:* `<question>`
*A:* `<mc answer 1>`
*B:* `<mc answer 2>`
*C:* `<mc answer 3>`
*D:* `<mc answer 4>`
*Answer:*

**mMMLU**

*Given the following question and answer choices, output the letter corresponding to the correct answer.*
*Question:* `<question>`
*A:* `<option a>`
*B:* `<option b>`
*C:* `<option c>`
*D:* `<option d>`
*Answer:*

**mHellaSwag**

*Given the following question and answer choices, output the letter corresponding to the correct answer.*
*Question:* `<ctx>`
*A:* `<ending 1>`
*B:* `<ending 2>`
*C:* `<ending 3>`
*D:* `<ending 4>`
*Answer:*

---

Table 3 lists the prompt templates used in our experiments reported in the main text.

# B   COMPUTE RESOURCES

We conducted experiments on models with fewer than 10B parameters using NVIDIA A100 GPUs (80GB) and on larger models using H100 GPUs. All models were able to fit on a single GPU for inference, and the experiments required approximately 200 hours of compute time.

# C   IMPACT OF LANGUAGE RELATIONS ON CROSS-LINGUAL TRANSFER

Figure 4 shows the pairwise positive transfer for all models on mMMLU. Like Ifergan et al. (2024), we find languages with known relations, such as Indic languages: Hindi (`hi`) and Bengali (`bn`); and Romance languages: Portuguese (`pt`), Spanish (`es`) and Italian (`it`); to be more closely connected and transfer more knowledge with each other. Indonesian (`id`) has higher rates of transfer with European languages, relative to Chinese (`zh`), Korean (`ko`) and Japanese (`ja`); indicating the role of shared scripts in knowledge transfer. The models are also less capable of generalizing predictions to and infer knowledge from low-resource languages: Swahili (`sw`) and Yoruba (`yo`). Knowledge learned in other languages remains inaccessible in `yo` as the models improve with increasing capacity.

# D   ACCURACY IN THE LATENT SPACE

Figure 5 shows full results of latent accuracy across languages.

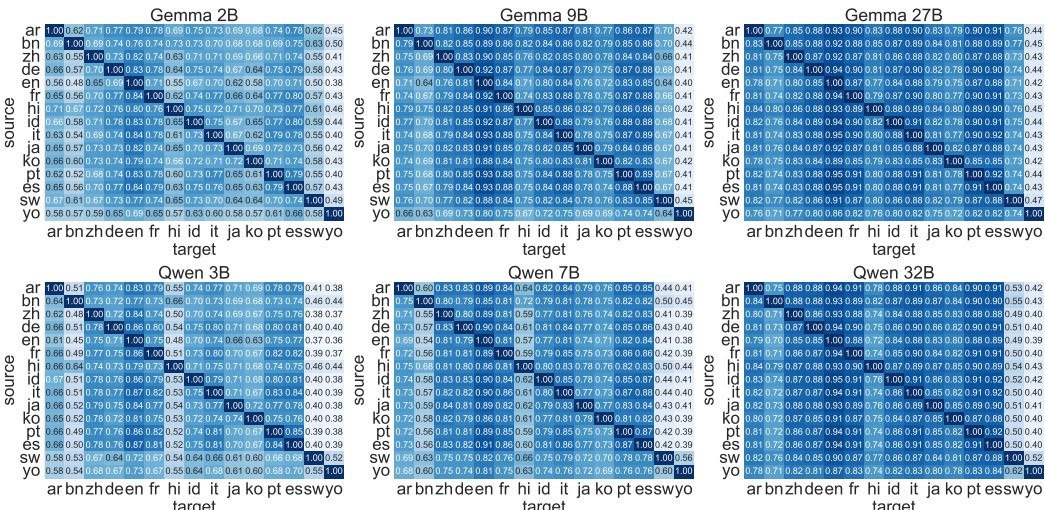

Figure 4: Positive transfer in all language pairs on mMMLU. Languages that are similar or are written in the same script are more likely to transfer with one another.

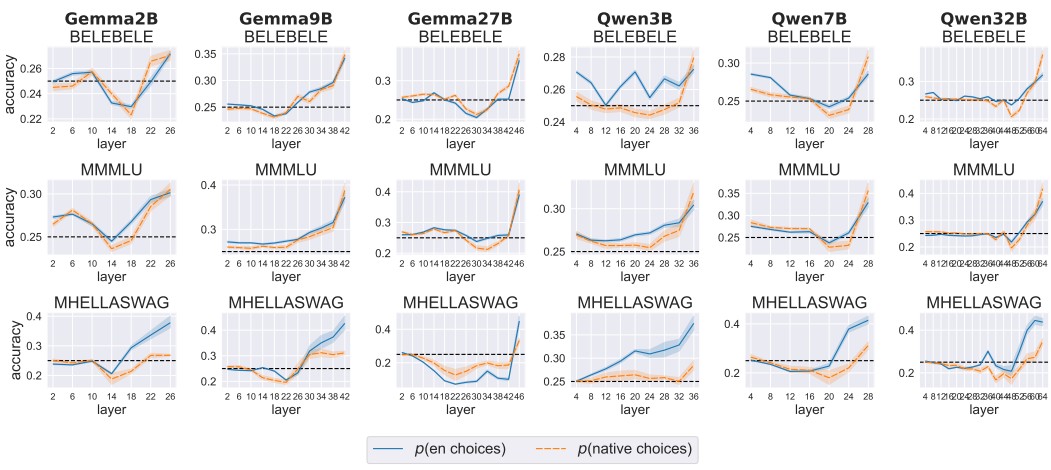

Figure 5: Full results of mean accuracy in the latent space across languages, analogous to Figure 2b. Chance-level accuracy is marked with a dashed line.

## E  PCA VISUALIZATION OF HIDDEN REPRESENTATIONS

Figure 6 provides a visual comparison of the hidden activations of 50 parallel BELEBELE questions between Gemma 9B and 27B, where the dimension of the vectors is reduced with the projection of Principal Component Analysis (PCA) (Wold et al., 1987). PCA is useful to assess if the hidden representations are linearly separable, which predetermines the efficacy of steering vectors (Rimsky et al., 2024). While the activations can be grouped by languages in the early layers in Gemma 9B (Figure 6a), they are mostly aligned between layers 18 and 38, and are separated by their written scripts (Latin vs. non-Latin) in the final layer. We note however that activations of low-resource languages, such as Yoruba (blue squares), are the last to be mapped and continue to be disjoint from the other languages across hidden layers.

In contrast, the latent space of Gemma 27B is more expansive, and the activations of the same languages are partially aligned with the others between layers 14 and 26, then become more tightly clustered throughout the layers. This confirms our previous analysis that larger models tend to operate in independent subspaces. Based on these observations, we expect steering vectors described in

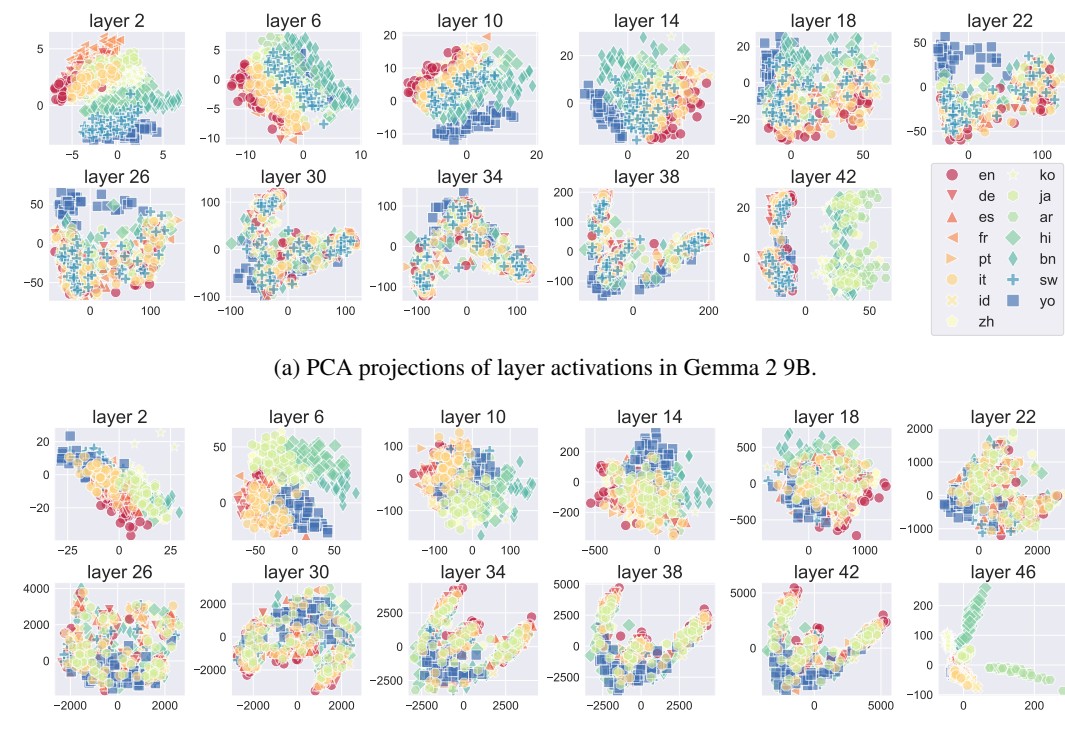

(a) PCA projections of layer activations in Gemma 2 9B.

(b) PCA projections of layer activations in Gemma 2 27B.

Figure 6: Layer activations with 50 parallel questions from BELEBELE.

Section 2.3 to be most effective in the middle layers, where the hidden activations across languages better align, indicating the use of shared semantic space in the process.

Figure 7 provides a comparison of hidden activations between Qwen 3B and 32B models. Like Gemma, the hidden representation in the middle layers is more aligned in the smaller model than the larger model.

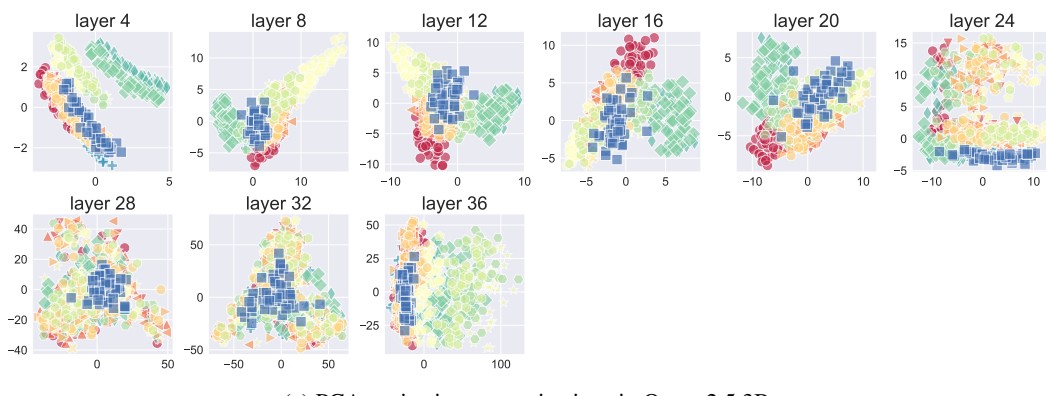

(a) PCA projections on activations in Qwen 2.5 3B.

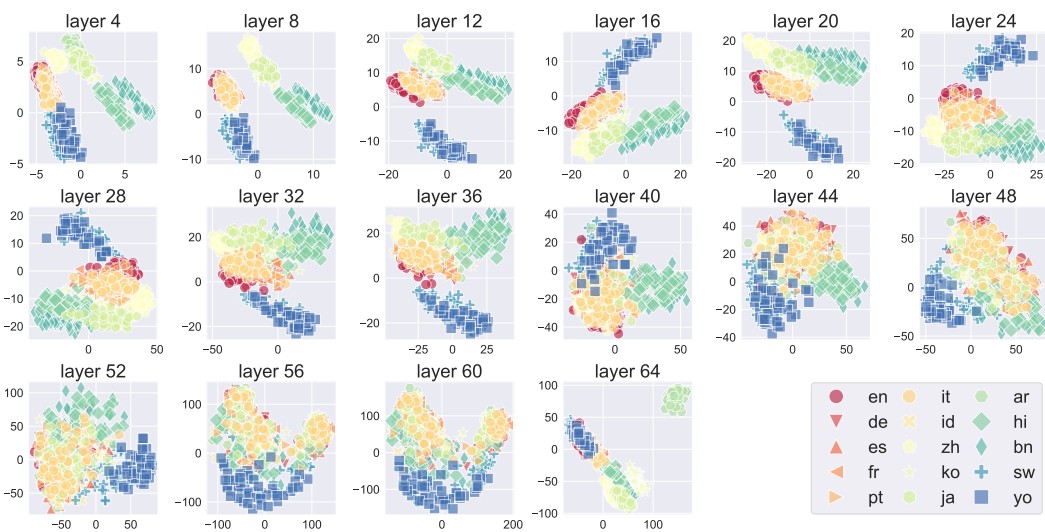

(b) PCA projections on activations in Qwen 2.5 32B.

Figure 7: Activations in Qwen models on 50 parallel questions from BELEBELE, analogous to Figure 6.

# F    ADDITIONAL RESULTS ON CROSS-LINGUAL ACTIVATION STEERING

## F.1    VARYING MULTIPLIER ON STEERED QWEN MODELS

We perform a sweep of the steering vector multiplier, $\gamma$, on Qwen 3B and 32B, and report the results in Figure 8.

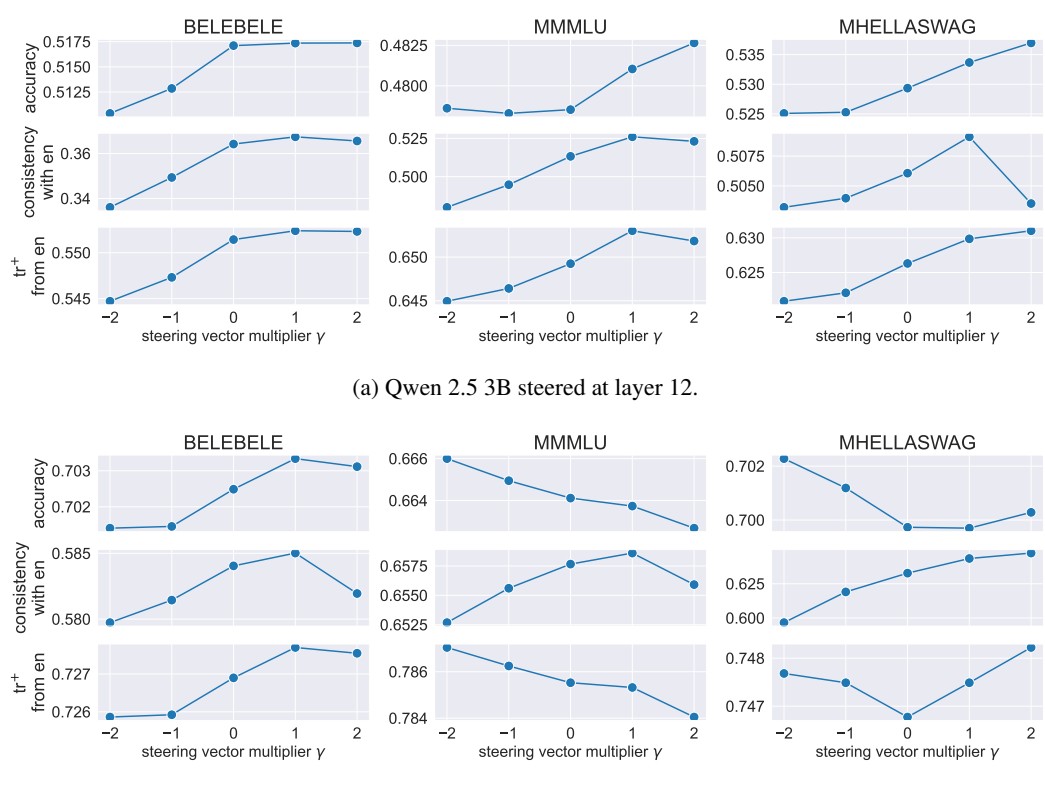

(a) Qwen 2.5 3B steered at layer 12.

(b) Qwen 2.5 32B steered at layer 12.

Figure 8: Qwen 3B steered towards latent English processing is more accurate, more consistent with English and receive more positive transfer from English. The experiment on Qwen 32B with mMMLU and mHellaSwag, however, reveal potential improved consistency at the expense of accuracy, particularly for models with large enough capacity to store information independently from English.

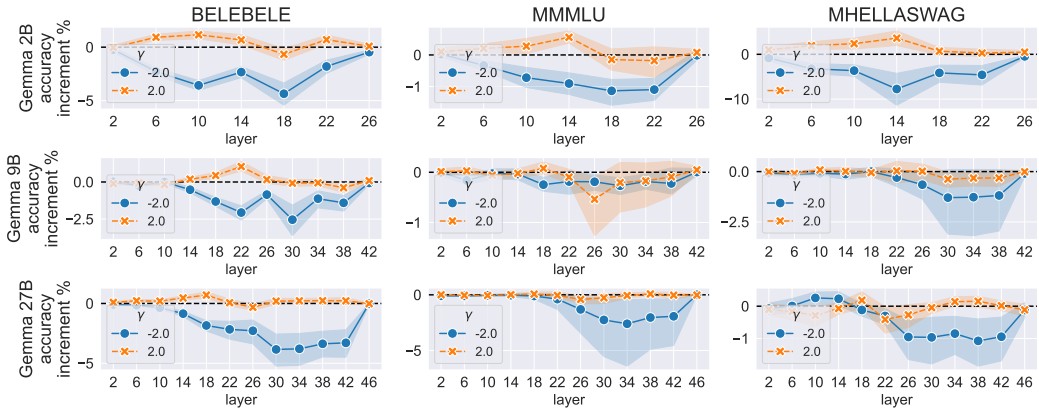

Figure 9: Gemma 2 models steered across layers, where $\gamma \in \{2.0, -2.0\}$. Shaded areas are 95% confidence intervals over languages.

## F.2 STEERING GEMMA MODELS ACROSS LAYERS

Steering is most effective in the middle layers, as evidenced by Figure 9.

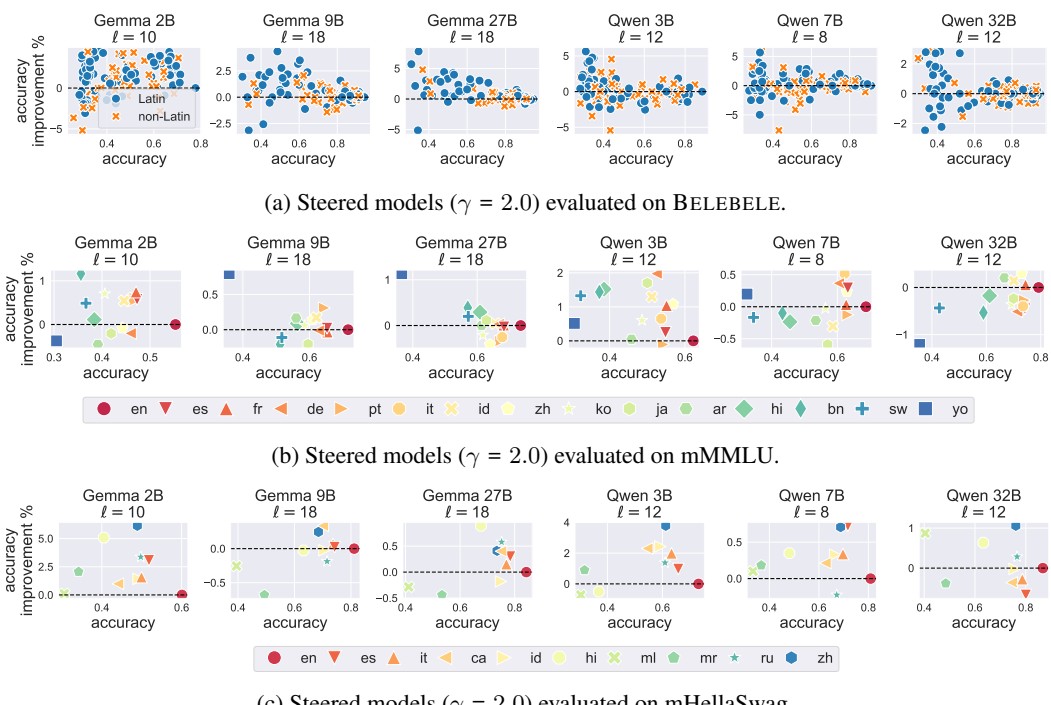

(a) Steered models ($\gamma = 2.0$) evaluated on BELEBELE.

(b) Steered models ($\gamma = 2.0$) evaluated on mMMLU.

(c) Steered models ($\gamma = 2.0$) evaluated on mHellaSwag.

Figure 10: Steering towards English is more effective on smaller models. Steering layers are shown at the top of the panels. The x-axes indicate the accuracy of non-steered models.

### F.3 ACCURACY, POSITIVE TRANSFER AND CONSISTENCY CHANGES IN STEERED MODELS ACROSS TASKS.

We show the changes in positive transfer from English and consistency with English in Figures 11 and 12 respectively. Overall, the results demonstrate that cross-lingual steering is effective on smaller models; whereas it is more difficult to induce stable and consistent transfer in larger models with simple activation addition.

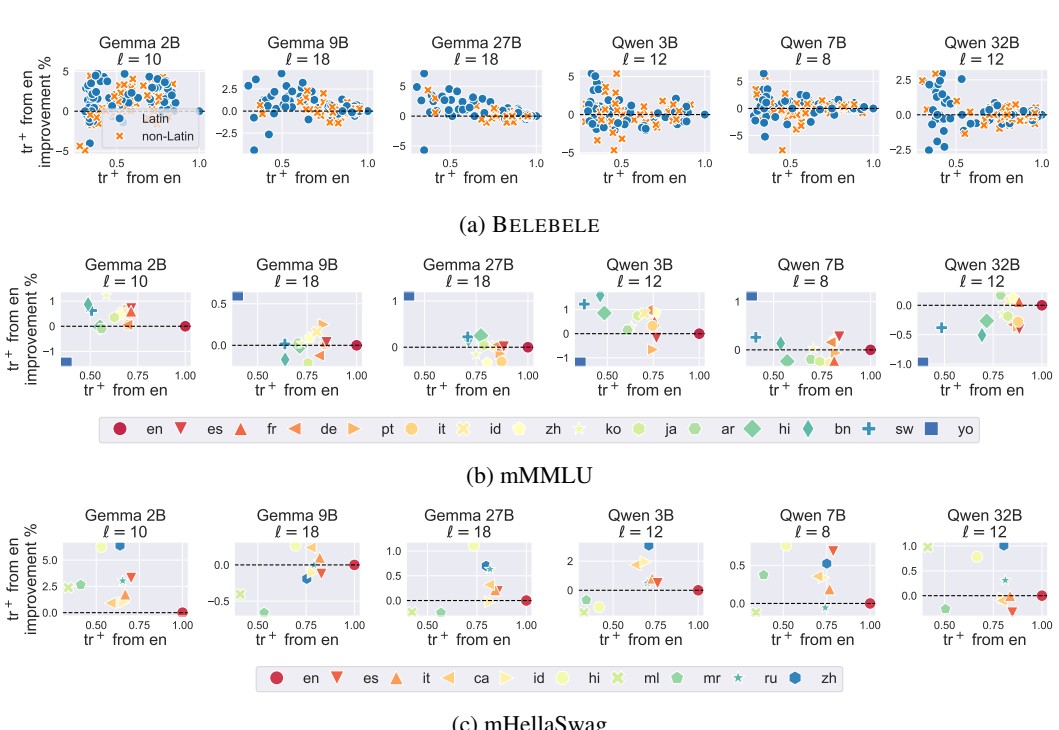

(a) BELEBELE

(b) mMMLU

(c) mHellaSwag

Figure 11: Changes in positive transfer from English in steered models ($\gamma = 2.0$) across tasks.

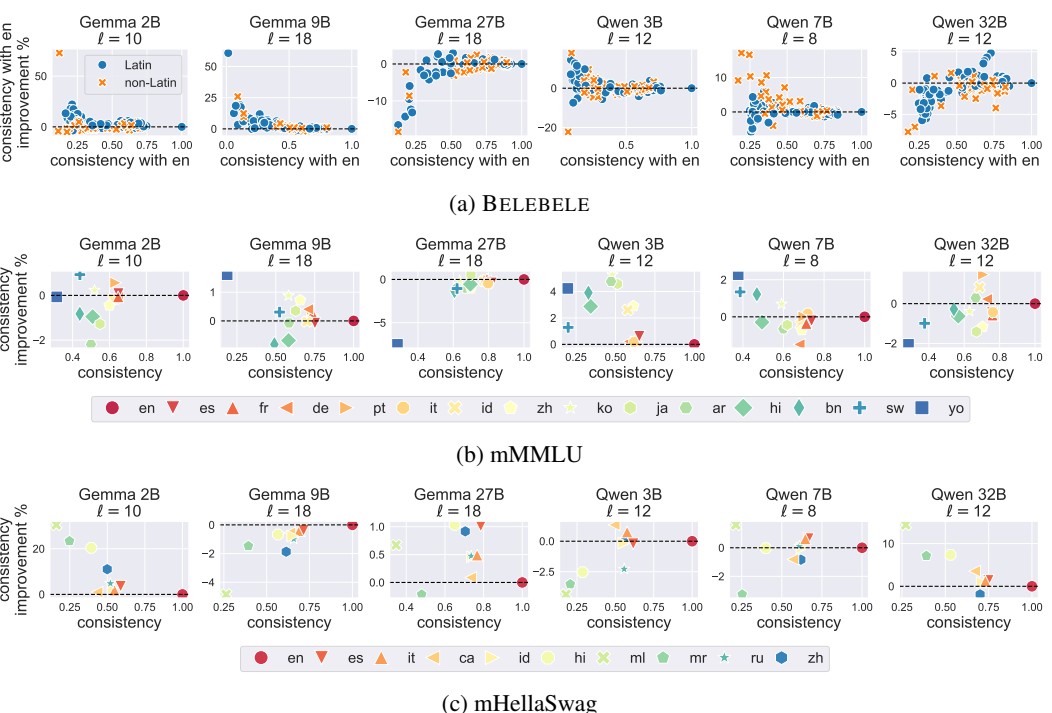

(a) BELEBELE

(b) mMMLU

(c) mHellaSwag

Figure 12: Changes in consistency with English in steered models ($\gamma = 2.0$) across tasks.

