# OpenReview forum: "Language-Specific Latent Process Hinders Cross-Lingual Performance"
_ICLR.cc/2026/Conference — Submitted to ICLR 2026_

### Official Review · Reviewer_v76E · 2025-10-27

**Soundness:** 3
**Presentation:** 4
**Contribution:** 2
**Rating:** 6
**Confidence:** 4

**Summary:**

The paper investigates why LLMs show inconsistent behavior when handling the same tasks across different languages. The authors analyze the internal representations of models such as Gemma 2 and Qwen 2.5 using techniques like centered kernel alignment (CKA), cosine similarity, and logit lens. They find that although larger models achieve higher multilingual accuracy, their internal representations become increasingly language-specific rather than shared, causing inconsistent reasoning across languages. Smaller models rely more on a shared, English-centric semantic space that supports more stable cross-lingual knowledge transfer.

To address this, the authors propose cross-lingual activation steering, a technique that nudges model activations toward English-aligned latent representations. This intervention improves multilingual reasoning and consistency in smaller models by enhancing their use of shared semantic structures. The work introduces a framework for quantifying cross-lingual alignment and offers an explanation for how model scale affects multilingual reasoning.

**Strengths:**

- Introduced cross-lingual activation steering to enhance reasoning performance.
- Combined CKA, cosine similarity, and the logit lens for multilingual analysis.
- Demonstrated a correlation between representation similarity and multilingual accuracy.

**Weaknesses:**

- The study relied primarily on multiple-choice tasks, which limited generalization to open-ended reasoning.
- Benefits are skewed toward languages written in Latin script.
- The validity of the findings is limited to two model families.
- There is no runtime or latency analysis conducted for steering.

**Questions:**

- Did you test how the steering method scales for larger or more complex models?
- How sensitive are results to the selection of layers?
- Could multilingual prompting (e.g., CoT in the target language) close the gap similarly?
- Did you try to construct shared-space vectors for non-English languages?

---

> ### Author Response · Authors · 2025-11-19
> **We will replicate our experiments on generative tasks, perform steering towards non-English languages, and test if multilingual prompting improve cross-lingual alignment.**
>
> We appreciate Reviewer v76E’s feedback and address them as follows.
>
> W1: Reviewer oQ2d raised similar concerns, and we repeat our response here. Parallel to our work, [1] learn weight matrices that minimize differences in representation between English and other languages. Their experiments show the learnable vectors to be slightly more competitive than our method (termed CAA in the paper), but the results demonstrate that steering vectors are effective across discriminative and generative tasks. Nonetheless, it is unclear if the methods could work beyond 7B/8B models and non-reasoning tasks. We will add experiments on open-ended math tasks (e.g., MGSM) in the revision of current work.
>
> W2: We agree that linear steering towards English benefits languages written in Latin scripts. It is however possible for similar steering towards a high-resource, non-Latin language (e.g., Arabic) to aid performance of lower-resourced languages of the same script (e.g., Urdu and Arabic transcriptions). We will add this analysis if space permits in the revision.
>
> W3: Gemma and Qwen models represent the SOTA multilingual models at the time of writing. We maintain that the experiments on two model families are sufficient to demonstrate our findings, in line with common practice in this field.
>
> W4: We perform linear steering that adds a single vector to a token at a single layer, this does not incur additional computational complexity in theory and in practice.
>
> Q1: We provide steering results of all models across datasets in Appendix F.3, in Figures 10, 11 and 12. Overall, linear steering is more effective on smaller models. However, it is more difficult to induce stable and consistent improvement on larger models with the same method.
>
> Q2: We demonstrate steering across layers in Figure 9, Appendix F.2. We find steering in the middle layers is the most effective.
>
> Q3: Thanks for raising this. It is possible that multilingual prompting helps align hidden representations in the target language, improving transfer and consistency. We will run exploratory analyses on this in the revision.
>
> Q4: This relates to W2. We will add analyses on languages written in major scripts (e.g., Arabic and Devanagari) to further support our hypotheses in the revised manuscript.
>
> [1] Wang et al., (2025) Bridging the Language Gaps in Large Language Models with Inference-Time Cross-Lingual Intervention

---

> ### Comment · Reviewer_v76E · 2025-11-27
>
> Thank you for your response. I appreciate your attention to the comments. I'll keep my score as I think it’s already high enough.

---

### Official Review · Reviewer_oQ2d · 2025-11-01

**Soundness:** 2
**Presentation:** 3
**Contribution:** 2
**Rating:** 4
**Confidence:** 3

**Summary:**

This paper investigates how LLMs process multilingual inputs and they often fail to maintain consistent reasoning across languages with a novel framework of repurposing CKA as a metric to study structure similarity between language representations. The paper studies the latent representations of models such as Gemma2, and Qwen2.5 across scales using multiple benchmark datasets.

**Strengths:**

- Build a framework for analysing cross-lingual transfer with well-selected metrics such as CKA, cosine-similarity and logit-lens to quantify language representation overlap, which provides a clear and interpretable way to study cross lingual transfer.
- The analysis across models and layers are comprehensive.
- The paper reframes multilingual reasoning as a latent-space alignment problem, providing a clear direction for multilingual output consistency.

**Weaknesses:**

- The task is completely limited to multi-choice reasoning questions. This is a clever choice that is easy to measure the cross-lingual transfer with unambiguous labels. However, it also limits the generalisation to open-ended and generative multilingual reasoning.
- “Humans have an innate ability to apply common knowledge and perform reasoning skills consistently across different languages” itself is a very contentious claim. This paper also doesn’t need Jerry Fodor’s nativism as motivation.
- As shown in Figure4, there is a huge disparity between transfer effects among languages that are not very related, such as from Arabic to Swahili. The averaged performances illustrated in the main body of the paper is obviously inflated. Without modelling the language relatedness in the analysis, it is difficult to assess the generalisability of the findings in the paper. At least the specific language’s relatedness with English.
- Also, in terms of language relatedness, the paper investigates steering vectors only towards English. The motivation seems to be that the training data is primarily English. On that note, the training data volume per language is also not taken into consideration at all. Various factors that can be impactful to the results are not considered.
- Overall, the paper is novel in repurposing a useful metric such as CKA for cross-lingual analysis, however, the study remains Anglocentric and linguistically shallow.

**Questions:**

- Consider language relatedness, and training data volume per language, instead of simply taking average results across languages, which can inflate the results because of higher resource latin script languages.

---

> ### Author Response · Authors · 2025-11-19
> **We will replicate our experiments on generative tasks and explore the impact of resource level, script sharing and language relation in the revision.**
>
> We thank Reviewer oQ2d for their constructive feedback and address their comments below.
>
> W1: Parallel to our work, [1] learn weight matrices that minimize differences in representation between English and other languages. Their experiments show the learnable vectors to be slightly more competitive than our method (termed CAA in the paper), but the results demonstrate that steering vectors are effective across discriminative and generative tasks. Nonetheless, it is unclear if the methods could work beyond 7B/8B models and non-reasoning tasks. We will add experiments on open-ended math tasks (e.g., MGSM) in the revision of current work.
>
> W2: A vast amount of research in cognitive science supports the existence of “core knowledge” and math reasoning skills in humans that are independent of language competency [2,3,4, inter alia]. However, we agree that Jerry Fodor’s claim can be extraneous for this argument. We will rephrase this in the revised manuscript.
>
> W3-5 & Q1:  Thanks for raising that we should account for influences such as training data and language relations on cross-lingual hidden state alignment, consistency and knowledge transfer. To a degree, our analysis in Figure 4 indicates low-resource languages receive little transfer from other languages, and the impact of language relation and script sharing on transfer. We agree that more robust analyses are required to further show if lower-resourced languages and languages that are distant from English perform poorly due to dissimilarity in representations. We will provide evidence on the importance of linguistic resources and language relations in the revision.
>
> [1] Wang et al., (2025) Bridging the Language Gaps in Large Language Models with Inference-Time Cross-Lingual Intervention
> [2] Strickland, B. (2017). Language reflects “core” cognition: A new theory about the origin of cross‐linguistic regularities. Cognitive science, 41(1), 70-101.
> [3] Dehaene, S. (2001). Précis of the number sense. Mind & language, 16(1), 16-36.
> [4] Piazza, M., & Dehaene, S. (2004). From number neurons to mental arithmetic: The cognitive neuroscience of number sense. The cognitive neurosciences, 3rd edition, ed. MS Gazzaniga, 865-77.

---

### Official Review · Reviewer_LmGN · 2025-11-01

**Soundness:** 3
**Presentation:** 3
**Contribution:** 3
**Rating:** 8
**Confidence:** 3

**Summary:**

The paper highlights the lack of consistency of outputs across languages, despite LLMs being capable of cross-lingual transfer. To measure generalization across languages, the paper measures representation similarity between languages by centered kernel alignment (CKA) and cosine similarity. They introduce three evaluation metrics to measure performance across languages: consistency, positive transfer and negative transfer.Their analysis show that LLMs don't use a shared semantic space which leads to low performance.They finally use steering vector to do steering towards English and find that it’s effective for smaller models.

**Strengths:**

1. The paper provides useful insights on how knowledge is represented and shared internally across languages in LLMs.  The authors investigate how LLMs transfer knowledge across languages. Through their experiments, they establish the usefulness of a shared semantic space for cross-lingual transfer.
2. The paper proposes a cross lingual steering approach to improve cross lingual transfer for smaller models.
3. Evaluation method is robust: The authors use ranking order for MCQ-styled questions across languages to measure consistency. They also measure positive and negative transfer.

**Weaknesses:**

1. Steering evaluation can include cross-dataset generalization to strengthen the claims. The current results only include effects on the same dataset.

**Questions:**

1. Do you notice any trends in consistency scores based on the linguistic distance of the language from English?
2. Does adding a steering vector towards English deteriorate performance on cultural/region-specific questions?

---

> ### Author Response · Authors · 2025-11-19
> **Thanks for recognizing the contribution of our work. We will add experiments on steering towards other languages.**
>
> We thank Reviewer LmGN for their appreciation of our work.
>
> W1: We provide steering results across mMMLU, Belebele and mHellaswag on Gemma 2B in Figure 3. In Appendix F.3, we show further evaluations on larger models and Qwen models with Figures 10, 11 and 12. Overall, cross-lingual steering is effective on smaller models. It is however more difficult to induce stable and consistent transfer in larger models with linear steering.
>
> Q1: In Appendix C we reveal the influence of linguistic relations and shared scripts in positive transfer (Figure 4). We observe more positive transfer between Indic languages (hi, bn) and Romance languages (es, pt, it). Indonesian has higher transfer rates with European languages than zh, ja, ko, indicating the role of shared written scripts.
>
> Q2: Thanks for raising this question. We will further investigate steering on INCLUDE [1], another MCQ datasets containing region/cultural specific questions and report our findings in the revised manuscript.
>
> [1] Romanou, A., Foroutan, N., Sotnikova, A., Nelaturu, S. H., Singh, S., Maheshwary, R., ... & Bosselut, A. INCLUDE: Evaluating Multilingual Language Understanding with Regional Knowledge. In The Thirteenth International Conference on Learning Representations.

---

### Official Review · Reviewer_y8RG · 2025-11-06

**Soundness:** 2
**Presentation:** 3
**Contribution:** 1
**Rating:** 2
**Confidence:** 4

**Summary:**

This paper studies cross-lingual consistency for the output when inputting the same query in different languages. The authors consider three factors: knowledge transfer (positive or negative), representation similarity, and activation steering. Then, the authors examine both small and large models, offering the key finding that a large model tends to handle each language independently, and a small model pushes all languages to a shared space.

**Strengths:**

Cross-lingual consistency is a recent topic. It evaluates the fairness in LLMs, which is very important in reall applications.

**Weaknesses:**

There are some concerns.

1.	Existing work [1] overshadows the novelty and contribution of this paper. For example, this paper follows a similar experimental design to [1], including CKA and Logits Lens examinations, layer-wise analysis, and activation steering.

2.	Experiments are limited, which makes the paper not conclusive. The authors only conducted experiments on multiple-choice datasets. How about generation tasks?

3.	While the paper is clear, the authors spend too many spaces on introducing existing works, e.g., CKA, Logits Lens, and methods for activation steering. I would like to see more in-depth analyses, which make the paper more fruitful.

[1] Are Knowledge and Reference in Multilingual Language Models Cross-Lingually Consistent?   https://arxiv.org/abs/2507.12838

**Questions:**

Refer to Weaknesses.

---

> ### Author Response · Authors · 2025-11-19
> **Our contribution is significantly different from [1]. More analyses will be added in the revised manuscript.**
>
> Thank you for your feedback and reference to a parallel work.
>
> W1:
> Our work is similar to [1] in
> 1) Demonstrating activation-based causal intervention to induce cross lingual consistency with English
> 2) Examining cross-lingual similarity by computing CKA between representations
> 3) Using logit lens to interpret hidden layers of multilingual models
>
> However, unlike [1]
> 1) Our measures of consistency and representational similarity do not rely on code-switched data, which potentially bias hidden representation towards English, inflating the cross-lingual scores.
> 2) Combined with CKA analysis, our use of logit lens successfully shows larger models operate in individual language subspace, explaining the disparity between multilingual performance and transfer capability.
> 3) We also provide strong evidence that similarity in hidden representation is predictive of cross-lingual consistency.
>
> We therefore argue that the contribution of our work is sufficiently different from [1], who further analysed the influence of vocabulary expansion, script overlap and multilingual learning objectives on consistency. We acknowledge their contribution and will cite their work in our revised manuscript.
>
> W2:
> Other reviewers raised similar concerns. We repeat our response here. Parallel to our work, [1] learn weight matrices that minimize differences in representation between English and other languages. Their experiments show the learnable vectors to be slightly more competitive than our method (termed CAA in the paper), but the results demonstrate that steering vectors are effective across discriminative and generative tasks. Nonetheless, it is unclear if the methods could work beyond 7B/8B models and non-reasoning tasks. We will add experiments on open-ended math tasks (e.g., MGSM) in the revision of current work.
>
> W3:
> Thank you for pointing this out. We will further summarize our method description and include the following analyses in the revision, following suggestions from other reviewers:
> 1) We will investigate if steering towards English is detrimental towards cultural/language-specific tasks (e.g., INCLUDE, [2]).
> 2) We will replicate our cross-lingual measures on open-ended math tasks (e.g., MGSM).
> 3) Like [1], we will analyse the influence of shared scripts, linguistic relations and resource level on cross-lingual knowledge sharing.
> 4) We will perform linear steering towards other high-resource languages (e.g., ar, hi) written in popular scripts (e.g., Arabic, Devanagari) and observe if this benefits languages related to the high-resource languages.
> 5) We will investigate if multilingual prompting further improve representational similarity in the latent space.
>
> [2] Romanou, A., Foroutan, N., Sotnikova, A., Nelaturu, S. H., Singh, S., Maheshwary, R., ... & Bosselut, A. INCLUDE: Evaluating Multilingual Language Understanding with Regional Knowledge. In The Thirteenth International Conference on Learning Representations.

---

> > ### Comment · Reviewer_y8RG · 2025-11-26
> > **Thanks for your feedback.**
> >
> > 1. Addressed.
> >
> > 2. and 3.:  I would like to raise the score to 4 at this stage for the current revision. There are still many tasks required, and I'm not sure the next revision has no additional concerns.

---

### Meta-Review · Area_Chair_1mBf · 2026-01-04

**Summary:**

As clearly stated in the title, this paper analyzes cross-lingual language models and finds that the reliance on individual language representations.

The major concerns raised by the reviewers include:
1. The contribution is unclear or incremental compared to existing literature (y8RG).
2. Reviewers have raised concerns about the experimental designs limited to multi-choice QA tasks (y8RG, oQ2d, v76E).
3. Reviewers have suggested a few additional experiments or edits, such as steering (LmGN) and clarification on language use (oQ2d).

**Reviewer Concerns:**

1. While the authors have claimed that there are significant differences between their work and the one brought out by Reviewer y8RG, I am not fully convinced on this point. Getting rid of code-switching data does not justify the claim of novelty.
2. The authors have promised to add MGSM (multilingual math QA) experiments, but did not update the manuscript accordingly.
3. Other minor concerns are addressed.

**Reviewer Scores:**

Reviewer y8RG has raised their rating from 2 to 4, but this does not cross the decision boundary.
The two reviewers with positive ratings are unlikely to change their ratings, and I guess Reviewer oQ2d will not change their rating since the promised experiments are not added.

---

### Decision · Program_Chairs · 2026-01-26

Reject